# A Study of Improved Two-Stage Dual-Conv Coordinate Attention Model for Sound Event Detection and Localization

**DOI:** 10.3390/s24165336

**Published:** 2024-08-18

**Authors:** Guorong Chen, Yuan Yu, Yuan Qiao, Junliang Yang, Chongling Du, Zhang Qian, Xiao Huang

**Affiliations:** 1School of Intelligent Technology and Engineering, Chongqing University of Science and Technology, No. 20, Daxuecheng East Road, Shapingba District, Chongqing 401331, Chinaqqzhang@cqust.edu.cn (Z.Q.); 2Department of Computing, The Hong Kong Polytechnic University, Hong Kong SAR 999077, China; xiaohuang@comp.polyu.edu.hk

**Keywords:** sound event detection and localization, sound event detection, sound source localization, coordinate attention

## Abstract

Sound Event Detection and Localization (SELD) is a comprehensive task that aims to solve the subtasks of Sound Event Detection (SED) and Sound Source Localization (SSL) simultaneously. The task of SELD lies in the need to solve both sound recognition and spatial localization problems, and different categories of sound events may overlap in time and space, making it more difficult for the model to distinguish between different events occurring at the same time and to locate the sound source. In this study, the Dual-conv Coordinate Attention Module (DCAM) combines dual convolutional blocks and Coordinate Attention, and based on this, the network architecture based on the two-stage strategy is improved to form the SELD-oriented Two-Stage Dual-conv Coordinate Attention Model (TDCAM) for SELD. TDCAM draws on the concepts of Visual Geometry Group (VGG) networks and Coordinate Attention to effectively capture critical local information by focusing on the coordinate space information of the feature map and dealing with the relationship between the feature map channels to enhance the feature selection capability of the model. To address the limitation of a single-layer Bi-directional Gated Recurrent Unit (Bi-GRU) in the two-stage network in terms of timing processing, we add to the structure of the two-layer Bi-GRU and introduce the data enhancement techniques of the frequency mask and time mask to improve the modeling and generalization ability of the model for timing features. Through experimental validation on the TAU Spatial Sound Events 2019 development dataset, our approach significantly improves the performance of SELD compared to the two-stage network baseline model. Furthermore, the effectiveness of DCAM and the two-layer Bi-GRU structure is confirmed by performing ablation experiments.

## 1. Introduction

SED is a technique designed to automatically identify and classify specific or multiple sound events (e.g., car horns, dog barks, or gunshots) from audio. In real life, acoustic scenarios are more complex, and sound events can occur simultaneously, i.e., sounds can overlap at the same time, for which the task is often referred to as polyphonic SED [1]. Initially, Kumar [2] and Mesaros [3] performed SED detection via Gaussian mixture models and Hidden Markov Models. With deep learning developments such as Gao [4], semi-supervised learning techniques are combined to solve the problem of fewer labeled datasets in the SED task to achieve semi-supervised Sound Event Detection. SSL is an important branch in the field of sound processing, whose core goal is to determine the location of the sound emitted. The delay of the audio signal emitted by the target to reach each microphone array element is not the same, so the SSL algorithm is used to process the audio signal from each array element to calculate the arrival direction of the sound source target, i.e., azimuth and elevation angles. For example, Time Difference of Arrival (TDOA) [5], Angle of Arrival (AOA) [6] and steered response power [7], etc., are commonly used methods and algorithms in SSL. Recently, Zhu [8] proposed a new single SSL model, the Icosahedral Feature Attention Network (IFAN), by applying deep learning techniques, which overcomes the limitation of environmental adaptability in generating features by traditional signal processing methods, and achieves target localization and tracking.

SELD combines SED and SSL with the aim of not only recognizing and classifying specific sound events in audio and detecting the time from the beginning to the end of an active event but also accurately determining the spatial location.

SELD is a technology with a wide range of applications, including areas such as surveillance systems [9], automated driving [10], wildlife monitoring [11], smart homes [12] and video conferencing automatic identification [13]. By integrating sound event recognition and spatial localization, SELD can provide richer and more detailed sound information that improves the perception and responsiveness of the environment for a wide range of machines and devices. SELD advances the use of audio analytics in several areas and is guaranteed to be able to handle complex sound scenarios.

With the rapid development of deep learning, the SELD field continues to benefit from it. Initially, Piczak [14] found that the convolutional neural network (CNN) was capable of capturing short-lived features in audio signals but was not suitable for dealing with overall time-series characteristics. In contrast, Parascandolo [15] found that Recurrent Neural Network (RNN) was good at exploiting time dependencies in audio sequences, as RNN can understand audio features that change over time, such as the persistence and variation of sounds. RNNs are able to remember previous information through their recurrent structure, which is suitable for sequential data. However, since each step of an RNN may depend on the output of the previous step, this structure makes it difficult to process in parallel, and problems such as difficulty in capturing the correlation between the beginning and the end of a sequence may be encountered when processing very long sequences. Adavanne [16] constructed a Convolutional Recurrent Neural Network (CRNN) including a convolutional module and a Bi-GRU module to train features for SED and Direction of Arrival (DOA), and named it SELDnet. The SELDnet combines the spatial feature extraction capabilities of a CNN with the time series processing advantages of an RNN. In particular, the Bi-GRU module, which can effectively deal with the before-and-after time dependency, is extremely crucial for accurately identifying and localizing sound events in continuous audio streams, and researchers have subsequently made continuous improvements on top of this model based on it.

Cao [17] proposed a two-stage strategy network for SELD, which first identifies the type of sound event in the SED branch, then uses the technique of transfer learning to extract DOA features, and finally predicts the Direction of Arrival of the sound event by training a SED mask. Although this approach is innovative in terms of strategy techniques and has achieved significant results, there are limitations in capturing local features in the convolutional layer in this network model. To better capture global and local features, Zhou [18] improved on Conformer and introduced the attention mechanism module, thus proposing the Dual Branch Attention Module (DBAM), which finally effectively detected two sound events occurring overlapping in the same time period in the Detection and Classification of Acoustic Scenes and Events (DCASE) 2020 task3 public dataset through a joint SELD network with soft parameter sharing.

In 2021, Hou [19] found that traditional channel attention mechanisms such as Squeeze-and-Excitation attention tended to ignore the more important positional information of the feature map, so they designed the Coordinate Attention mechanisms to pay special attention to feature map coordinate information. Unlike the traditional channel attention mechanism, Coordinate Attention compresses the feature tensor compressed into a single vector, dividing the channel attention into two 1D feature encoding processes, thus preserving the precise position information in the other direction. The module has shown excellent performance in tasks such as image classification, object detection and semantic segmentation. Wang [20] proposed the CA-Wav2Lip system based on Coordinate Attention to achieve the accurate synchronization of speech and lip movements in the field of video translation. Jiang [21] also designed a speech enhancement model CCAUNet by Coordinate Attention, which focuses on the time and frequency correlation of speech data and spatial location information at the same time. Inspired by Coordinate Attention and the SELD approach for the two-stage strategy, we propose a DCAM, which combines dual convolutional blocks and a Coordinate Attention module, based on the DCAM module in the two-stage strategy network model improvement, to propose a TDCAM for SELD. In this work, our contributions are as follows:A DCAM is proposed to effectively capture local features on the feature layer. The module uses dual convolutional blocks and the Coordinate Attention mechanism to effectively capture local features on the feature layer, and at the same time optimizes the information flow and feature fusion between the channels, enhances the feature selection ability of the model, and has a significant enhancement effect on improving the accuracy of SELD in complex environments.A SELD-oriented TDCAM incorporating DCAM is proposed to enhance the modeling capabilities of temporal features in complex scenarios. The model combines the proposed DCAM and is increased to a two-layer Bi-GRU in response to the limitation of using only one layer of Bi-GRU in the two-stage strategy model proposed by Cao [17]. The use of two layers of Bi-GRU can better capture the contextual information of the sequence data, and improve the modeling capability of the temporal features. The introduction of the frequency mask and time mask data enhancement techniques enriches the diversity of the training data and enhances the generalization performance of the model. The above two improvements effectively enhance the overall performance of the model.

In Section 2, the detailed structure of the TDCAM and the DCAM is described in detail. In Section 3, the experimental setup is given, including the datasets, experiment settings, and evaluation metrics. In Section 4, comparison experiments with other models and ablation experiments are shown and analyzed. In Section 5, the proposed method in this paper and the outlook for the future are concluded.

## 2. Methods

In this section, the details of the implementation of SELD-oriented TDCAM are described. Figure 1 illustrates the specific structure of the model, which extracts different features based on different formats of audio data. The network divides the traditional SELD network into the SED branch and DOA branch for SED and SSL, respectively, so that SELD achieves better results.

### 2.1. Feature Extract

The dataset used in this experiment is the publicly available dataset TAU Spatial Sound Events 2019 given in DCASE 2019 task3 [22], which consists of audio data in both First-Order Ambisonics (FOA) and Microphone Array (MIC) formats. Therefore, we extract Log-Mel Frequency Spectrum (Log-Mel) and intensity vector features for FOA-type audio data, and for MIC-type audio data, we extract Log-Mel and Generalized Cross-Correlation with Phase Transform (GCC-PHAT) features, respectively. The three different features are extracted as follows.

#### 2.1.1. Log-Mel

Log-Mel features are extracted for both FOA and MIC format audio files. The key advantage of Log-Mel features is their ability to mimic human auditory perceptual properties, which makes them an extremely effective feature in deep learning methods for analyzing audio data. The Log-Mel feature extraction method starts by performing a Short-Time Fourier Transform (STFT), which results in a *S* complex matrix containing the magnitude and phase information of the frequency components, denoted as the magnitude in the time–frequency domain, and is expressed as equation:(1)$$S\left(k,m\right)=\sum_{n=0}^{N-1}x\left(n\right)\omega \left(n-mH\right)e^{-j\frac{2\pi kn}{N}}$$
where $x\left(n\right)$ is the audio signal, ω is the window function, *N* is the FFT size, *H* is the hop size, *k* is the frequency index, and *m* is the time index.

Subsequently, the absolute value of the STFT squared is calculated to obtain the power spectral density, which is used to describe the energy distribution of the signal at each frequency with the following equation:(2)$$P\left(f,t\right)={\left|S\left(f,t\right)\right|}^{2}$$
where $\left|S\left(f,t\right)\right|$ is the modulus of the STFT complex result corresponding to frequency *f* and time *t*.

Next, the FFT spectrum is mapped to Mel frequencies, a step that is accomplished by applying the Mel filter matrix *M* to the power spectral density *P*. The Mel filter matrix *M* is a transformation matrix from linear frequencies to Mel frequencies, with each filter covering a specific range of frequencies that are nonlinearly distributed on the Mel scale. The formula is expressed as:(3)$$S_{mel}=M\cdot P$$

Finally, a logarithmic conversion of the Mel spectrum is performed to simulate the non-linear perception of sound by the human ear. The Log-Mel spectrum $S_{LogMel}$ is calculated using the following equation:(4)$$S_{LogMel}=10\cdot \log_{10}\left(\max\left(S_{mel},a_{\min}\right)\right)$$
where $a_{\min}$ is a very small positive number used to avoid the problem of calculating logarithms with a zero denominator.

After processing all the channels, the Log-Mel features obtained for each channel are superimposed to form the final feature array.

#### 2.1.2. Intensity Vector

For audio data in the FOA format, we also extract its intensity vector feature, which is an important analysis method for FOA audio to extract spatial information and is widely used in Virtual Reality, Augmented Reality, 3D Audio Rendering, etc. [23]. FOA audio usually contains four components W,X,Y,Z, where *W* is usually the omnidirectional pressure component, and X,Y,Z is the velocity component pointing to different spatial directions in the Cartesian coordinate system. Extracting the intensity vector features, we firstly STFT the signals of all channels to obtain the frequency domain representation:(5)$$P_{i}=S\left(i\right)$$
where *i* indicates a different channel W,X,Y,Z, and i=1,2,3 corresponds to X,Y,*Z*, respectively.

Next, the intensity components in the three directions are computed by calculating the frequency domain representation of the reference channel (usually, the *W* channel is used) with the signals in the other three directions $I_{1},I_{2},I_{3}$. The intensity is calculated by multiplying the complex conjugate of the reference signal with the signals in each direction and then taking the real part, which is expressed by the formula:(6)$$I_{i}=\mathrm{Real}\left(\bar{P_{\mathrm{ref}\;}}\cdot P_{i}\right)$$
where $\bar{P_{\mathrm{ref}\;}}$ is the complex conjugate of the reference signal, and $P_{i}$ is the STFT result of the X,Y,Z axis signal.

In order to eliminate the effect of the magnitude of the intensity vector and focus only on its direction, the intensity vector needs to be normalized. First, the mode of the intensity vector is calculated:(7)$$n=\sqrt{I_{1}^{2}+I_{2}^{2}+I_{3}^{2}}$$

The intensity components in each direction are then divided by this mode:(8)$$I_{i}^{\prime }=\frac{I_{i}}{n}$$

This yields a unit intensity vector with each of its components in the range [−1,1].

Then, just like Log-Mel feature extraction, the normalized intensity components are mapped to Mel frequencies using the Mel filter matrix *M*:(9)$$I_{mel,i}=M\cdot I_{i}^{\prime }$$

This step converts each normalized intensity component $I_{i}^{\prime }$ into a representation at the Mel frequency.

Finally, the sets of Mel frequency intensity vectors for the three directions are stacked into one array, the array of intensity vectors for that FOA audio signal.

#### 2.1.3. GCC-PHAT

For audio data in the MIC format, we also extract its GCC-PHAT feature, which is a widely used technique for sound source localization and is particularly suitable for estimating the time delay difference TDOA between two microphone signals. It optimizes the accuracy of the time delay estimation mainly by calculating the signal’s cross-correlation function and using the phase information [24]. The extracted GCC-PHAT feature method starts with separate audio channels $x_{\mathrm{sig}}$ and $x_{\mathrm{refsig}}$ extracted from multichannel audio data, and STFT is performed on the two signals to obtain their spectral representations $X\left(f,t\right)$ and $Y\left(f,t\right)$:(10)$$\begin{array}{cc} X(f,t) & =\mathrm{STFT}\{x_{\mathrm{sig}}\}\\ Y(f,t) & =\mathrm{STFT}\{x_{\mathrm{refsig}}\} \end{array}$$

Next, the reciprocal power spectrum of the two signals is calculated, which is performed by taking the product of the spectrum of one signal conjugated to the spectrum of the other:(11)$$R\left(f,t\right)=X\left(f,t\right)\cdot \bar{Y(f,t)}$$
where $\bar{Y(f,t)}$ is denoted as the conjugate form of Y(f,t).

The subsequent application of the phase transform aims to normalize the amplitude of the reciprocal power spectrum, retaining only the phase information, thus removing the effect of signal amplitude and improving the robustness of the estimation:(12)$$R_{\mathrm{PHAT}}\left(f,t\right)=\frac{R\left(f,t\right)}{\left|R\left(f,t\right)\right|}$$

Finally, the processed spectrum $R_{\mathrm{PHAT}}\left(f,t\right)$ is converted back to the time domain by Inverse Fast Fourier Transform (IFFT) to obtain the cross-correlation function $r_{\mathrm{GCC}-\mathrm{PHAT}}$:(13)$$r_{\mathrm{GCC}-\mathrm{PHAT}}\left(t\right)=\mathrm{IFFT}\left\{R_{\mathrm{PHAT}}\left(f,t\right)\right\}$$

In order to correctly resolve the delays obtained from the cross-correlations and to move the negative time delays to the front of the array, thus simplifying the search for the location of the peaks, it is necessary to circularly shift the results of the calculations:(14)rGCC−PHAT=CrGCC−PHAT−mm22:,rGCC−PHAT:mm22
where C is the concatenate operation, and *m* is the number of Mel filters.

In this paper, two feature stacks, Log-Mel and GCC-PHAT, are used as input features for MIC audio data, where the time lag range included in the GCC-PHAT spectrum must be −mm22,mm22.

### 2.2. Network Architecture

The overall architecture of the network proposed in this paper is shown in Figure 1, which divides the SELD task into two parts of the SED branch and DOA branch but uses different structures at the feature layer and the Bi-GRU layer from the two-stage network [17]. In the training process, features of shape C×T×F are first extracted from FOA (7 channels) or MIC (10 channels), where *C* is the number of channels of the feature, *T* is the number of frames of the data, and *F* is the number of Mel filters or the number of GCC-PHAT delay samples.

These extracted features are first passed through the feature layer, which consists of four DCAM modules. Each DCAM module contains two consecutive convolutional blocks, a Coordinate Attention layer and a 2×2 average pooling layer. Through the four DCAM modules, the *C* dimension of the input data is raised to 64, 128, 256, and 512 channels in turn. Then, the data pass through the global average pooling layer to subtract the *F* dimension so that the subsequent two-way GRU layer can accept the input. In contrast to the two-stage network [17], this paper found that a separate Bi-GRU layer does not effectively capture the timing information in the audio data, so two Bi-GRU layers are used to capture the information more efficiently. Finally, the output layer feeds data into a fully connected layer of size *K*, where *K* is the number of event types. The final SED branch prediction keeps the input data *T* dimension constant by applying a Sigmoid activation function on the time dimension *T* and upsampling according to the activation threshold.

The DOA branch is trained next, where the DCAM parameters in the feature layer are transferred from the SED prediction branch and fine-tuned. The parameters of the subsequent Bi-GRU and output layers are initialized. The output of the fully connected layer of the DOA branch is a vector of K×2 linear values, where the *K* events correspond to the azimuth and elevation angles, respectively. During the training process, the SED Ground Truth value is used to mask the angle of the model output. This means that only outputs that are relevant to the actual sound source location are retained, while irrelevant outputs are suppressed. In this way, the model is only affected by angles that are relevant to the actual sound source position during training, leading to the better learning of correct localization information.

In the inference stage, the computation of the SED branch is first performed to obtain the prediction results for acoustic event detection. Then, these prediction results are used as SED masks for the orientation prediction of DOA branches. The advantage of this is that each branch can focus on solving its own task without interference by other tasks. At the same time, the DOA branch can still benefit from the information from the SED branch to improve its performance. This network structure will improve the performance of the whole system while maximizing the correlation between different tasks.

### 2.3. Dual-Conv Coordinate Attention Module

In the evolution of deep learning, the VGG network [25] has been regarded as one of the classical deep convolutional neural network models. Its core idea is to use multiple convolutional kernels of a small size and deeper network structure to improve the model performance. This idea has widely inspired and influenced the field of deep learning. Although previous studies, such as Cao [17], cleverly applied the similar VGG core idea to SELDnet and achieved remarkable results, it also implies that the method has some limitations in terms of local feature perception capability. Coordinate Attention, as a spatial attention module, demonstrates excellent capability in capturing feature information. It calculates the importance of each position of the feature map based on its coordinate information, and thus decides how much attention should be paid to the features at a particular position. Therefore, it can help the model to not only better understand the spatial structure and local features of the image but also effectively capture the correlation between the channels in the process of feature representation, which provides richer information for the model’s learning and inference, and thus improves the model’s performance and generalization ability.

Inspired by VGG net and Coordinate Attention, we construct the DCAM module to replace the two-stage network convolutional layers. As shown in Figure 2, each DCAM module consists of several components. First, there are two consecutive 2D convolutional blocks, using a structure similar to VGG networks, each containing multiple 3×3 sized convolutional kernels with 1×1 step size and 1×1 padding size, followed by a batch normalization layer and a ReLu activation function. Next is an improved Coordinate Attention module for enhanced spatial attention feature extraction. Finally, a 2×2 average pooling layer is used to reduce the spatial dimensionality of the feature map.

Overall, the DCAM module adds more convolutional layers compared to those in the traditional SELDnet and two-stage network, improving the depth and feature representation of the model, while the application of the Coordinate Attention module helps the whole model to better understand the spatial structure and features of sound events.

Figure 3 shows the detailed steps of the Coordinate Attention module. For a tensor $X\in R^{C\times H\times W}$, the coordinate information is first embedded, i.e., average pooling is performed along the horizontal and vertical directions of X to obtain $X_{avg}^{x}\in R^{C\times H\times 1}$ and $X_{avg}^{y}\in R^{C\times 1\times W}$, respectively.

Next, $X_{avg}^{x}$ and $X_{avg}^{y}$ are converted to the same plane and stitched together to obtain $X_{avg}^{x}\in R^{C\times (W+H)\times 1}$. Next, by convolving the kernel into a 1×1 Conv block and batch normal layer, with Coordinate Attention [19] using the Sigmoid activation function, we use the ReLu activation function to transform the features into intermediate feature mapping $F\in R^{C^{\prime }\times \left(H+W\right)\times 1}$, which encodes the features in the spatial horizontal direction and spatial vertical direction.

This intermediate feature is then partitioned into two tensors $F_{x}\in R^{C^{\prime }\times H\times 1}$ and $F_{y}\in R^{C^{\prime }\times W\times 1}$, which are transformed with different 1×1 convolution blocks and Sigmoid activation functions to obtain the weights $W_{x}\in R^{C\times H\times 1}$ and $W_{y}\in R^{C\times 1\times W}$ for the horizontal and vertical directions of the input tensor X. Finally, the weights are multiplied with the tensor X to obtain the output Y, denoted as:(15)$$Y=X⊙W_{x}⊙W_{y}$$

This Coordinate Attention module takes into account both inter-channel relationships and positional information, and it not only captures cross-channel information but is also able to focus on the feature map local information, which allows the model to more accurately localize to the recognition target area.

### 2.4. Data Augmentation

Specaugment [26], a powerful data augmentation technique, has gained significant attention in audio signal processing and automatic speech recognition. The frequency mask and time mask are two commonly used data enhancement techniques. This study, on the other hand, helps the model to better process and understand real-world audio data by using both frequency mask and time mask data augmentation on the spectrogram of the features, in order to improve the robustness and generalization of the model. As shown in Figure 4, applying a frequency mask on the frequency dimension of the audio achieves data enhancement by randomly selecting a portion of the frequency region in the spectrogram and replacing its value with zero or mean. Similarly, applying a time mask on the time dimension of the audio achieves data enhancement by randomly selecting a portion of the time region in the spectrogram and replacing its value with zero or the mean. Finally, a combination of these two methods is used to vary both the frequency and time dimensions simultaneously, which will effectively enhance the model’s adaptability and robustness to different audio environments.

### 2.5. Loss Function

For the SELD task in this study, the SED branch predicts the categories of multiple sound events, which can be seen as a multi-label classification problem. Therefore, in the SED branch, we use Cross-Entropy (CE) [27] as the loss function for the SED task, which is calculated as follows:(16)$$loss_{SED}=-\frac{1}{N}\sum_{n=1}^{N}\sum_{t=1}^{T}y_{nt}\times \log\left(p\left({\hat{y}}_{nt}\right)\right)$$
where $y_{nt}$ and ${\hat{y}}_{nt}$ are denoted as the reference and predicted values of the sound category probability for the *t*-th frame of the *n* sound event in the audio, respectively.

As for the DOA branch, to predict the sound event localization, it can be regarded as a problem of the multi-output regression task. Therefore, in the DOA branch, we adopt the Mean Absolute Error (MAE) as the calculation of the loss function for the DOA task, which is calculated as follows:(17)$$loss_{elevation/azimuth}=\frac{\sum_{t=1}^{T}\left|y_{t}^{\prime }-{{\hat{y}}^{\prime }}_{t}\right|\times m_{t}}{v}$$
where $y_{t}^{\prime }$ and ${{\hat{y}}^{\prime }}_{t}$ are the reference and predicted values of the azimuth or elevation of the sound event for the *t*-th frame of the sound event in the audio, respectively. $m_{t}$ is the mask value of the *t*-th frame, which is used to filter the parts that do not need to be considered. *v* is the normalized value, i.e., the number of elements in the mask with a value of 1, used to normalize the sum of the errors. This loss function is mainly used as a loss for the calculation of azimuth or elevation angles, while the loss function $loss_{DOA}$ for the whole DOA task is expressed as the sum of $loss_{elevation}$ and $loss_{azimuth}$.

And for the whole SELD network, we use the joint loss function, which is the combination of SED branch loss and DOA branch loss. And we set γ weights to perform joint optimization to obtain more accurate classification and localization performance, which is formulated as follows:(18)$$loss_{SELD}=\gamma loss_{DOA}+\left(1-\gamma \right)loss_{SED}$$
where γ is a variable parameter, set to 0.2 in this study.

## 3. Experiment

### 3.1. Datasets

The publicly available dataset TAU Spatial Sound Events 2019 (Ambisonic and Microphone Array), a development dataset given by DCASE 2019 task3, was used for this study [22]. This dataset consists of two development datasets including TAU Spatial Sound Events 2019 Ambisonic, and TAU Spatial Sound Events 2019 Microphone Array. TAU Spatial Sound Events 2019 Ambisonic was recorded by a first-order high-fidelity stereo First-Order Ambisonics device, whereas TAU Spatial Sound Events 2019 Microphone Array was recorded by a tetrahedral microphone array device. Therefore, in this study, different features are extracted for the data recorded by the two different devices. For FOA-type data, we extract Log Mel features and 3-channel acoustic Intensity Vectors features, while for MIC-type data, we extract Log Mel features and GCC-PHAT features.

The TAU Spatial Sound Events 2019 Development dataset contains 400 audio files of 1 min in length, with a sampling rate of 48,000 Hz and a channel count of 4 channels. The audio is synthesized from spatial room Impulse Responses (IRs) that simulate sound propagation in five room locations. There are a total of 11 spatialized sound event categories, such as knock, drawer and clear throat, with a maximum of two overlapping events in the audio. As shown in Table 1 these files are divided into four groups of 100 each for cross-validating the performance of the model, where these splits consist of audio recordings and corresponding metadata describing the sound events and their corresponding locations in each recording. Each audio file has 504 different combinations of azimuth, elevation, and distance. These combinations precisely define the location of the sound source in space, thus enabling the dataset to support training and testing for accurate SELD in variable environments. Finally, in order to create realistic sound recordings, natural ambient noise collected at the IR recording location is added to the synthesized recordings to give an average SNR of 30 dB for sound events.

### 3.2. Training Setup

First, in this study, the training set is divided into every 2 s as a segment (with complementary zeros filled in for audio data that cannot be divided into whole lengths), where every 1 s segment is 100 frames, while the validation and test sets are not divided. The data batch sizes for training are set to 64 (i.e., 64 segmented 2 s audio segments as a batch), and the max epoch for training is set to 80 so that the total number of iterations for each fold is the number of all the audio in the folds after the total duration is segmented. Second, the short-time Fourier transform SFFT is performed on the initial audio data by setting a 1024-point Hann window with a 320-point frame shift. The experiments are conducted to build the model using the PyCharm platform in Python version 3.8 and PyTorch version 2.3.0. Training is performed on a CPU model Intel i9-13900K and GPU model RTX 4090 with driver version 552.22 and cuda version 12.1. The Adam [28] optimizer is used during training, setting the initial learning rate of the Adam optimizer to 0.001 and adjusting the learning rate by multiplying it by 0.9 at intervals of every 2000 iterations when the total number of iterations is greater than 23,000. Finally, a threshold of 0.5 is set for SED as a binarization of the predictions.

### 3.3. Evaluation Metrics

In this study, the assessment metrics provided by the DCASE 2019 Challenge task3 are used. In targeting the SED branch [29], the SED error rate and F-Score evaluation metrics are used, while for the DOA branch [30], the DOA error rate and frame recall evaluation metrics are used. We aggregate the number of True Positives (TP), False Positives (FP), and False Negatives (FN) for the whole data and calculate the metrics based on the overall values.

The SED error rate in a SED branch is measured by insertions (*I*), deletions (*D*), and substitutions (*S*). In a segment *k*, I(k) refers to instances where inactive events are incorrectly reported as active, often seen as redundant system-generated active events. D(k) denotes active events that are under-reported as inactive, where these events do occur but are treated as non-existent in the system output. S(k) involves active events being misreported as other types of active events, where the erroneous event is still active but does not align with the correct event. This follows the equation:(19)$$\begin{array}{cc} I(k) & =\max\left(0,\mathrm{FP}(k)-\mathrm{FN}(k)\right)\\ D(k) & =\max\left(0,\mathrm{FN}(k)-\mathrm{FP}(k)\right)\\ S(k) & =\min\left(\mathrm{FN}(k),\mathrm{FP}(k)\right) \end{array}$$

The SED error rate is calculated from the total number of segments *K* and the number of active sound events in segment *k*. A smaller error rate in the SED branch indicates a better performance of the SED system, which is given by the formulation:(20)$$ER_{SED}=\frac{\sum_{k=1}^{K}I\left(k\right)+\sum_{k=1}^{K}D\left(k\right)+\sum_{k=1}^{K}S\left(k\right)}{\sum_{k=1}^{K}N\left(k\right)}$$

In deep learning classification tasks, most studies use Precision (*P*) and Recall (*R*) to measure the *F*-score, and generally, the higher the *F*-score, the better the performance of the SED system. Where *P* indicates the proportion of samples predicted by the model to be positive classes that are positive classes, and *R* denotes the proportion of samples that are positive classes that are correctly predicted by the model to be positive classes. Both are defined by the equation:(21)$$\begin{array}{cc} P & =\frac{TP}{TP+FP}\\ R & =\frac{TP}{TP+FN} \end{array}$$ Therefore, the *F*-score calculated based on *P* and *R* in the SED branch in this study can be equal:(22)$$F=\frac{2\cdot P\cdot R}{P+R}$$
or it can also be represented by TP, FP and FN:(23)$$F=\frac{2\cdot TP}{2\cdot TP+FP+FN}$$ For the DOA branch, we use two evaluation metrics: DOA error rate and frame recall. The DOA error rate represents the average angular error between the predicted DOA and the reference DOA when calculated in degrees. The smaller the error value, the closer the predicted sound source direction is to the actual direction, and the better the performance of the system. For an audio segment with a total time-frame length of *T*, the DOA prediction and reference list for one of its time-frames *t* are $DOA_{R}^{t}$ and $DOA_{E}^{t}$, respectively, and then the DOA error rate is defined as:(24)$$ER_{DOA}=\frac{1}{\sum_{t=1}^{T}D_{E}^{t}}\sum_{t=1}^{T}H\left(DOA_{R}^{t},DOA_{E}^{t}\right)$$
where $D_{E}^{t}$ is the number of DOAs in $DOA_{E}^{t}$ at the *t*-th frame, and H is the Hungarian algorithm used to solve the assignment problem.

The Hungarian algorithm [31] is commonly used to solve allocation problems, such as matching predicted and reference DOAs in multi-target tracking or sound source localization. By estimating the cost between each pair of predicted and reference DOAs calculated based on the angular distance between them, the least costly match is sought for each pair. This efficient pairing ensures that the overall error is minimized, which can be expressed by the following equation:(25)$$\sigma =\arccos\left(\sin\lambda_{E}\sin\lambda_{R}+\cos\lambda_{E}\cos\lambda_{R}\cos\left(\left|\phi_{R}-\phi_{E}\right|\right)\right)$$
where, for the reference DOA, the azimuth and elevation angles are denoted as $\phi_{R}\in \left[-\pi ,\pi \right)$ and $\lambda_{R}\in \left[-\pi ,\pi \right)$, respectively, and similarly for the predicted DOA, the azimuth and elevation angles are denoted as $\phi_{E}\in \left[-\pi ,\pi \right)$ and $\lambda_{E}\in \left[-\pi ,\pi \right)$, respectively.

Second, we evaluate how many of the reference DOAs are correctly identified by the predicted DOAs using the frame recall metric. This metric is particularly useful in scenarios where the number of predicted DOAs does not align with the actual number of reference DOAs, such as when multiple source directions exist within a single time frame but only some are predicted by the model. A higher frame recall value indicates a better match between the number of DOA predictions and the actual number of reference directions, reflecting improved performance of the DOA system. The frame recall formula is given by:(26)$$FR=\frac{\sum_{t=1}^{T}1\left(D_{R}^{t}=D_{E}^{t}\right)}{T}$$
where $D_{R}^{t}$ represents the number of DOAs actually referenced in the *t*-th frame of $DOA_{R}^{t}$, and 1() is an indicator function. Specifically, if the condition $D_{R}^{t}=D_{E}^{t}$ is satisfied, it means that the number matches and the function results in 1. If they do not match, the function returns 0.

## 4. Results and Analysis

### 4.1. Baseline

The following will introduce the baseline models for comparison in the experiment, which comes from the outstanding team of DCASE 2019 Task 3.

**DCASE 2019 Task3 Baseline** [16]: The approach employs the SELD traditional CRNN model, with successive frames of the spectrogram as input. The network produces two outputs simultaneously: one for detecting sound events and the other for locating their source.

**Two-stage network** [17]: The method uses a two-stage approach to SED and Direction of Arrival Estimation (DOAE). The first stage is to train a model for SED, which recognizes sound events and their time information; the second stage is to use the features trained by SED to train DOAE, using the true labels of SED as the mask for training DOAE.

**Noh** [32]: The method employs a three-stage strategy for SELD. This includes Sound Activity Detection (SAD) using Multi-Resolution Cochlear Glectrograms (MRCG) and CRNN to determine event presence and count; SED utilizing Log-Mel spectrograms and CRNN for event identification; and DOA employing GCC-PHAT features and CNN to estimate sound source direction.

**Zhang** [33]: The method divides the SELD task into three steps. These include data augmentation, network prediction, and post-processing. In the data augmentation step, the SpecAugment [26] data augmentation method is used, and the network prediction then divides the CRNN network into a SED network and a DOA network. Finally, the post-processing step uses Prior Knowledge-based Regularization (PKR) to predict and locate events.

**Nguyen** [34]: This method is a two-step method to achieve SELD for FOA format datasets. The first step is to detect sound events using CRNN and estimate the sound source direction using the single-source histogram algorithm. The second step directly combines the results of SED and DOA, and uses rule logic to fuse the results of each time frame to improve the performance of SELD.

**Grondin** [35]: This method specifically focuses on utilizing the characteristics of the MIC datasets. Using the log-amplitude and phase spectrograms from the microphone array data and the GCC-PHAT features as input, the method implements SELD using two independent CRNNs. One CRNN performs SED, while the other estimates the TDOA for each microphone pair.

In order to fully validate the effectiveness of our approach and ensure that the models are compatible with the FOA and MIC formats in the TAU Spatial Sound Events 2019 Ambisonic and Microphone Array dataset, we conduct a series of comparative experiments using the same evaluation metrics of Section 3.3. Based on the FOA and MIC format datasets, we select the official baseline model for DCASE 2019 Task 3 [16] for comparison with several models that performed well in this challenge task, including the Cao [17], Noh [32], and Zhang [33] models. We also choose the models of Nguyen [34] and Grondin [35] for comparison based on these two different formats of data, respectively.

### 4.2. Comparative Result

In Table 2 and Table 3, the SELD performance for both MIC and FOA format datasets are significantly enhanced using the proposed method, respectively. Specifically, in terms of the SED metric, the proposed method outperforms several other models. While it may not be the top performer in the DOA metric, it markedly lowers the DOA error rate by 0.6 and 1.4 compared to the two-stage model for both datasets. Additionally, the frame recall metric also exhibits slight increases of 0.012 and 0.014, further validating the effectiveness and reliability of the proposed method in practical applications.

Furthermore, by incorporating data augmentation techniques into the proposed method, further improvements are observed across all performance metrics. The data-enhanced method demonstrates superior performance in both datasets, underscoring the critical role of frequency mask and time mask data augmentation techniques in enhancing model generalization and adapting to complex sound environments.

To more intuitively validate the SELD performance of the proposed method, we randomly select two test audio clips from the MIC format dataset. These clips come from different test cuts and environmental settings:

**split1_ir0_ov1_9**: This clip is from the first set of data (Fold 1), location number 0, which contains an overlapping sound event (1 overlapping sound event), which is the 9th audio clip.

**split2_ir3_ov2_75**: Tis clip is from the second set of data (Fold 2), location number 3 (location number 3), which contains two overlapping sound events (2 overlapping sound events). This is the 75th audio clip.

Figure 5 and Figure 6 illustrate the Sound Event Detection, along with the azimuth and elevation references and predictions for the two audio clips, respectively. The horizontal axis of each plot represents time in seconds, corresponding to the total duration of the audio clips. The vertical axis for the SED reference and prediction indicates the 10 audio categories, ranging from 0 to 10. The vertical axes for the azimuth and elevation references and predictions display angles in degrees.

Figure 5 shows the predictions for the test data **split1_ir0_ov1_9** according to the proposed method. In terms of SED, the model effectively predicts various sound event categories and generally matches the start and end times of the events accurately, demonstrating high precision. While the azimuth and elevation angle predictions show some deviations, they overall effectively indicate the sound source direction. Notably, the azimuth predictions are closer to the reference values, illustrating the model’s effectiveness in the horizontal orientation.

Figure 6 presents the results for the test data **split2_ir3_ov2_75**, which involves multiple overlapping sound events. Despite some errors in predicting specific categories in SED, the proposed method successfully recognizes most sound events and accurately predicts several of the event categories. Although there is some dispersion and deviation in the azimuth and elevation localization predictions, the model captures the general trend of the sound source direction well.

### 4.3. Ablation Experiments

To study the proposed method in detail, in this subsection, we set up the two-stage network [17] as the baseline model for the ablation experiments, followed by setting up the improvement to a two-layer Bi-GRU module, using the proposed DCAM module and using the data-enhanced model for the ablation experiments, respectively. These ablation experiments will further validate the effectiveness of the sub-modules of the method.

In Table 4 and Table 5, a series of ablation experiments performed on FOA and MIC datasets are shown, aiming at evaluating the contribution of the different components to the proposed method. The baseline model is the two-stage network, which shows relatively high benchmark performance on both datasets. By improving to a two-layer Bi-GRU, a noticeable improvement in DOA error rate metrics can be seen, showing the effectiveness of the two-layer Bi-GRU in improving the accuracy of the direction estimation. Also, the experiments using DCAM in both datasets show some performance improvement, especially in the F-score of SED, which demonstrates the usefulness of DCAM in improving the accuracy of SED.

In applying the combination of two-layer Bi-GRU and DCAM in the proposed method, we see further improvement in all the metrics. This combined approach not only reduces the SED error rate but also achieves the best performance in terms of DOA error rate and frame recall, indicating that the synergistic effect of the two techniques is significant in improving the overall performance of the model.

Finally, the performance of the model in both datasets is further improved by applying data enhancement to the proposed method. The significant improvement of each metric in both format datasets and these results fully demonstrate the importance of data enhancement in improving the robustness and accuracy of SELD systems.

## 5. Conclusions

In this paper, we propose a TDCAM model based on the DCAM module. We combine the VGG network idea and Coordinate Attention module to design and propose the DCAM module instead of the convolutional layer of SELDnet to capture the features. In addition, the TDCAM model introduces the proposed DCAM module and improves on the two-stage network proposed by Cao [17], which not only introduces a two-layer Bi-GRU layer to enhance the model’s ability to model time series but also employs the data enhancement techniques of frequency masking and time masking, which further improves the model’s performance. Experimental results on the TAU Spatial Sound Events 2019 development dataset demonstrate the effectiveness of the method, which shows better performance in SELD tasks compared to the official baseline model and several other models. The ablation experiments likewise validate the need for a two-layer Bi-GRU and DCAM module. Therefore, in future work, we plan to explore alternative time-series modeling methods to replace Bi-GRU and investigate other multi-task learning strategies with a view to further improving the accuracy and efficiency of SELD.

## Figures and Tables

**Figure 1 sensors-24-05336-f001:**
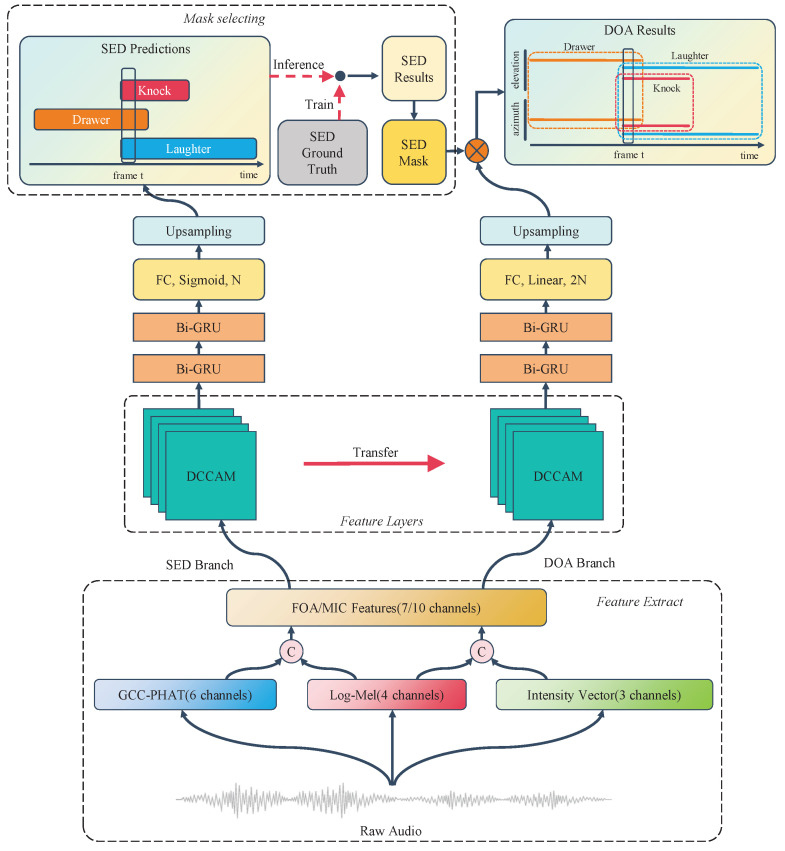
Overall architecture of the TDCAM network. The TDCAM comprises two primary branches: the SED branch for Sound Event Detection and the DOA branch for Direction-of-Arrival estimation. Both branches share a common feature extraction layer, yet utilize distinct network structures at higher levels for task-specific feature transformation and prediction.

**Figure 2 sensors-24-05336-f002:**
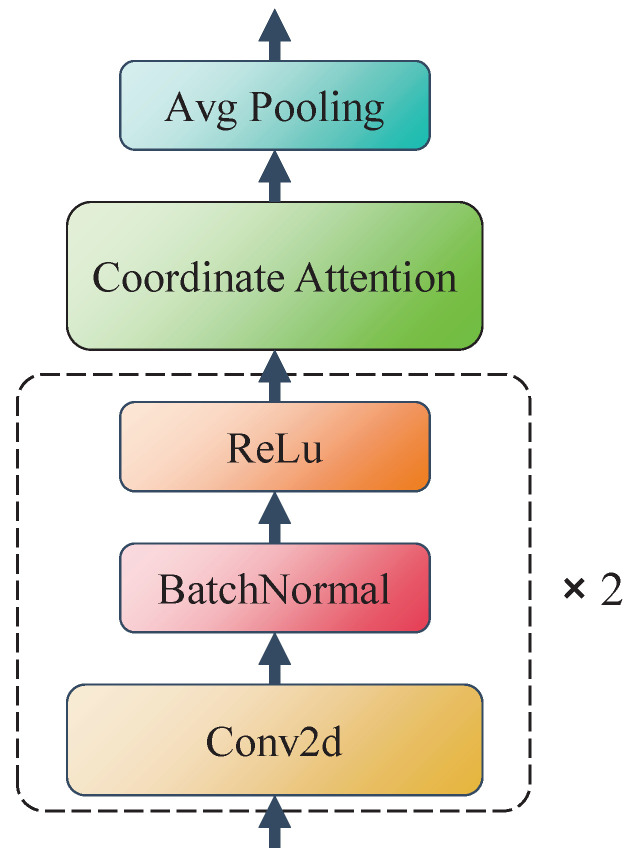
Overall architecture of the DCAM module.

**Figure 3 sensors-24-05336-f003:**
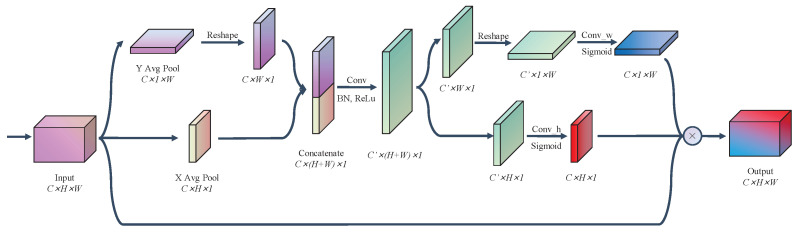
The detailed steps of the Coordinate Attention module. It acquires horizontal and vertical spatial attention features through pooling while simultaneously extracting channel attention features. The combination of these features is then fused with the input features to generate an attention-weighted output feature map.

**Figure 4 sensors-24-05336-f004:**
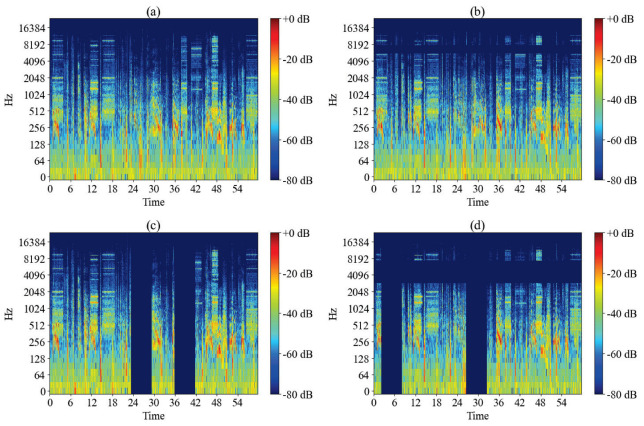
Data augmentation visualization. (**a**) The original audio spectrogram. (**b**) The spectrogram after applying frequency mask augmentation to the original. (**c**) The spectrogram after applying time mask augmentation to the original. (**d**) The spectrogram after applying both frequency mask and time mask augmentations to the original.

**Figure 5 sensors-24-05336-f005:**
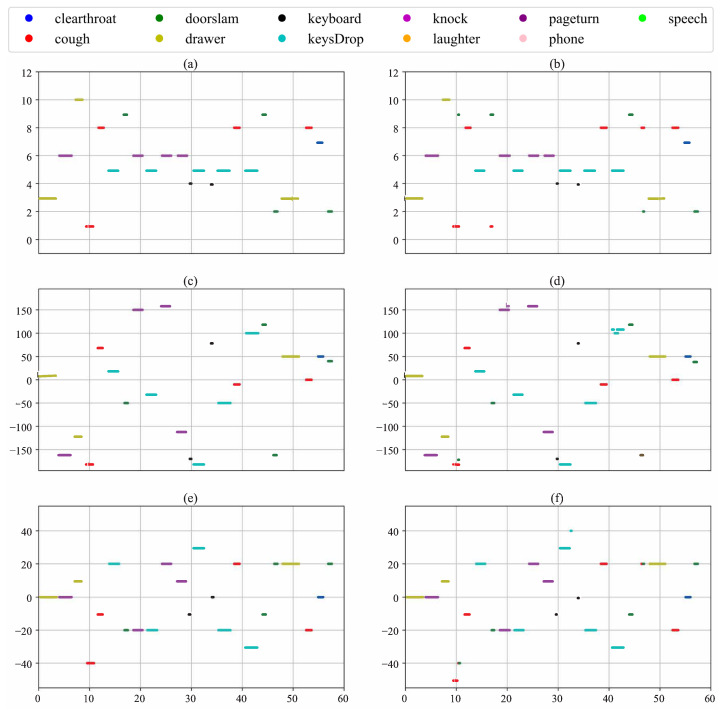
Visualizationof **split1_ir0_ov1_9** results for test data. (**a**,**b**) Visualization of these test data SED reference and predicted values, respectively. (**c**,**d**) Visualization of these test data DOA azimuth reference and predicted values, respectively. (**e**,**f**) The visualization of these test data DOA elevation angle reference and predicted values, respectively.

**Figure 6 sensors-24-05336-f006:**
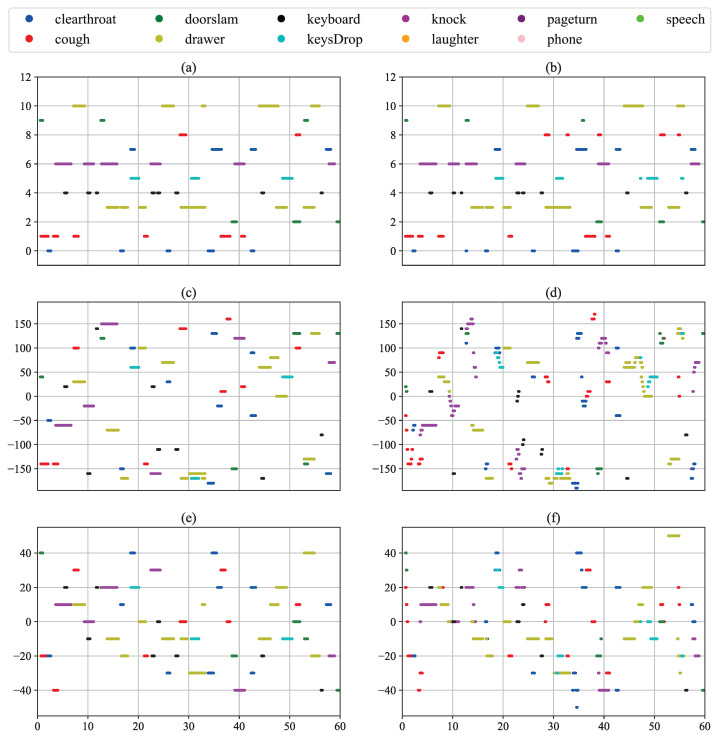
Visualization of **split2_ir3_ov2_75** results for test data. (**a**,**b**) The visualization of this test data SED reference and predicted values, respectively. (**c**,**d**) The visualization of these test data DOA azimuth reference and predicted values, respectively. (**e**,**f**) The visualization of these test data DOA elevation angle reference and predicted values, respectively.

**Table 1 sensors-24-05336-t001:** Grouping of the TAU Spatial Sound Events 2019 Development datasets.

Folds	Training Splits	Validation Split	Testing Split
Fold 1	3, 4	2	1
Fold 2	4, 1	3	2
Fold 3	1, 2	4	3
Fold 4	2, 3	1	4

**Table 2 sensors-24-05336-t002:** SELD performance of the proposed method and comparison model on the FOA format dataset.

Models	${ER}_{SED}↓$ *	F↑	${ER}_{DOA}↓$	FR↑
DCASE 2019 task3 [16]	0.35	0.800	30.8	0.840
Two-Stage network [17]	0.17	0.907	9.0	0.855
Noh [32]	0.25	0.850	10.6	**0.928 ****
Zhang [33]	**0.14**	0.916	24.8	0.908
Nguyen [34]	0.21	0.869	**5.1**	0.889
TDCAM(ours)	0.15	0.915	8.4	0.867
TDCAM(ours) + Data Augment	**0.14**	**0.919**	8.1	0.871

* In the table, the symbol ↓ indicates that a lower value is better for the metric, while the symbol ↑ signifies that a higher value is better. ** In the table, bolded performance metrics indicate that the model performs best on these indicators.

**Table 3 sensors-24-05336-t003:** SELD performance of the proposed method and comparison model on the MIC format dataset.

Models	${ER}_{SED}↓$ *	F↑	${ER}_{DOA}↓$	FR↑
DCASE 2019 task3 [16]	0.34	0.799	28.5	0.854
Two-Stage network [17]	0.15	0.909	10.0	0.860
Noh [32]	0.28	0.842	**4.0 ****	**0.918**
Zhang [33]	0.15	0.913	25.8	0.895
Grondin [35]	0.20	0.878	6.5	0.876
TDCAM(ours)	**0.14**	0.918	8.6	0.874
TDCAM(ours) + Data Augment	**0.14**	**0.920**	8.1	0.874

* In the table, the symbol ↓ indicates that a lower value is better for the metric, while the symbol ↑ signifies that a higher value is better. ** In the table, bolded performance metrics indicate that the model performs best on these indicators.

**Table 4 sensors-24-05336-t004:** Ablation experiments for FOA format audio dataset.

Models	${ER}_{SED}↓$*	F↑	${ER}_{DOA}↓$	FR↑
Two-Stage network (Baseline) [17]	0.17	0.907	9.0	0.855
use 2*Bi-GRU	0.16	0.908	7.7	0.862
use DCAM	0.16	0.910	8.8	0.860
TDCAM(2*Bi-GRU + DCAM)	0.15	0.915	8.4	0.867
TDCAM + Data Augment	**0.14 ****	**0.919**	**8.1**	**0.871**

* In the table, the symbol ↓ indicates that a lower value is better for the metric, while the symbol ↑ signifies that a higher value is better. ** In the table, bolded performance metrics indicate that the model performs best on these indicators.

**Table 5 sensors-24-05336-t005:** Ablation experiments for MIC format audio dataset.

Models	${ER}_{SED}↓$*	F↑	${ER}_{DOA}↓$	FR↑
Two-Stage network (Baseline) [17]	0.15	0.909	10.0	0.860
use 2*Bi-GRU	0.16	0.909	8.3	0.865
use DCAM	0.15	0.912	9.0	0.864
TDCAM(2*Bi-GRU + DCAM)	**0.14 ****	0.918	8.6	**0.874**
TDCAM + Data Augment	**0.14**	**0.920**	**8.1**	**0.874**

* In the table, the symbol ↓ indicates that a lower value is better for the metric, while the symbol ↑ signifies that a higher value is better. ** In the table, bolded performance metrics indicate that the model performs best on these indicators.

## Data Availability

The TAU Spatial Sound Events 2019 development dataset used in the experiments of this paper is available at https://dcase.community/challenge2019/task-sound-event-localization-and-detection#download (accessed on 1 April 2024).

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
