# Peer review of "A Study of Improved Two-Stage Dual-Conv Coordinate Attention Model for Sound Event Detection and Localization"

_sensors, 2024, doi:10.3390/s24165336_

Round 1

Reviewer 1 Report

Comments and Suggestions for Authors

The topic addressed in the paper is interesting, and the results are well presented and discussed. I think this work is worth publishing, and just a minor revision is required:

-          In the abstract, VGG is not defined. Bi-GRU is not defined in the abstract but later in the text. However, it should be also defined in the abstract.

-          There is a typo (question mark) at lines 107, 163, 413, 414.

-          Typo at line 191 (lowercase i).

-          Typo at line 298 (in).

-          Typo at line 341 (repeated words).

-          Typo at line 393: Recall (P).

-          Typo at line 396: P instead of R.

-          Typo at line 435: … datasets, we used…

-          In table 3, both results with 0.14 score should be highlighted in bold, similarly to Table 2.

-          I suggest rewriting the list of acronyms at the end of the manuscript in alphabetical order.

Comments on the Quality of English Language

The paper is not always easy to read, and the English could be improved. Here a couple of examples:

The delay of the audio signal emitted by the target to reach each microphone array element is not the same, then the SSL algorithm is used to…

What does “then” mean? The sentence should be revised.

Coordinate Attention by Hou[19] in 2021 as part of the IEEE/CFF Conference on Computer Vision and Pattern Recognition (CVPR). They found that…

The two sentences are not connected. E.g.: In 2021, Hou [19] used the Coordinate Attention model, finding that…

Reviewer 2 Report

Comments and Suggestions for Authors

In the paper the authors propose the application of the two-stage dual-conv Coordinate Attention model for the solution of the sound event detection and localization task. The peculiarity of the developed approach is the application of VGG network and Coordinate Attention module for capturing the features of different sound events. Verification of the proposed model was performed through comparative studies with other approaches developed for the considered problem, using the TAU Spatial Sound Events 2019 development dataset. The comparative study demonstrated the higher efficiency of the two-stage, dual-conv Coordinate Attention model compared to the existing approaches.

In general, the paper is well organized. All research methods, novelty of this study and obtained results were discussed in sufficient detail. However, there are some remarks that I think will improve the paper slightly:

1. The efficiency of the Two-Stage Dual-conv Coordinate Attention Model is verified by the comparison study with other models. It is worth to add additional details about the models used to confirm the good performance of the developed approach. Namely, the features of some models are not discussed in the paper and only references to the corresponding research have been left in the paper. Furthermore, another important detail is the implementation of all models considered in the study. Did the authors develop these models themselves using the data from the papers of other reseachers? Or are the models available online? How these models were trained and how were their efficiency metrics obtained? Which parameters of these models were used?

2. The format of Figures 5 and 6 does not meet the requirements of the Journal. If a figure consists of several sub-figures, each sub-figure should be identified by a letter. Each sub-figure should be described in the figure captions. Please consider the Journal's requirements and make appropriate changes.

3. As far as I can see, the paper contains several mistypes. There are mistypes in lines 107, 163, 413, 414. It seems to me that the sentence on line 297 is not well written.

4. In my opinion, the title of the paper will be better if you use the full name of the abbreviation "SELD".
